# Human Amniotic Membrane for the Treatment of Cryptoglandular Anal Fistulas

**DOI:** 10.3390/jcm11051350

**Published:** 2022-03-01

**Authors:** Carlo Ratto, Ornella Parolini, Angelo Alessandro Marra, Valentina Orticelli, Angelo Parello, Paola Campennì, Veronica De Simone, Diletta Trojan, Francesco Litta

**Affiliations:** 1Proctology Unit, Fondazione Policlinico Universitario Agostino Gemelli IRCCS, 00168 Rome, Italy; angeloalessandromarr@libero.it (A.A.M.); angelo.parello@proctocenter.it (A.P.); campennip@hotmail.it (P.C.); veronicadesimone@libero.it (V.D.S.); francesco.litta@policlinicogemelli.it (F.L.); 2Department of Medicine and Translational Surgery, Università Cattolica del Sacro Cuore, 00168 Rome, Italy; 3Fondazione Policlinico Universitario Agostino Gemelli IRCCS, 00168 Rome, Italy; ornella.parolini@unicatt.it (O.P.); valentina.orticelli@unicatt.it (V.O.); 4Department of Life Science and Public Health, Università Cattolica del Sacro Cuore, 00168 Rome, Italy; 5Tissue Bank, Fondazione Banca Dei Tessuti Di Treviso Onlus, 31100 Treviso, Italy; dtrojan@fbtv-treviso.org

**Keywords:** amniotic membrane, anal fistula, wound healing, regenerative medicine

## Abstract

Background: Implantation of the amniotic membrane and their derivatives can have a beneficial effect on tissue repair and regeneration. We report for the first time the implant of an amniotic membrane in a patient affected by cryptoglandular anal fistula. Methods: A patch of human amniotic membrane was implanted in a female patient affected by an anterior transphincteric fistula. Following an accurate curettage of the anal fistula, the cryopreserved amniotic membrane was thawed and then washed in the operating room; one side of the membrane was transfixed with a resorbable suture thus creating an implantable fusiform patch. The membrane was subsequently implanted into the fistula tract from the external to the internal opening. The inner and outer parts of the membrane were then sutured to the internal and external fistula openings. Results: No intraoperative or postoperative complications occurred. The patient was discharged one day after the procedure after an uneventful hospitalization. At the 1-week, 1- and 3-month follow-up visits no pain (VAS 0) was referred by the patient and no inflammation was evident at the level of the previous external fistula opening. Conclusions: The implant of human amniotic membrane in a patient affected by cryptoglandular anal fistula was safely and easily performed. Moreover, future studies to assess the efficacy in the long-term follow-up are needed.

## 1. Introduction

Surgical treatment of anal fistulas (AFs) represents an ongoing challenge in proctology [1].

The lack of certainty regarding the etiology of AFs could partly clarify the persistence of controversies surrounding this topic. A therapeutic approach based on etiological principles, rather than on AF morphology and characteristics of the chosen surgical technique, has been advocated [2]. Evidence suggested that epithelialization of the AF tract and persistence of chronic inflammation may play a decisive role in the persistence and recurrence of disease [3,4,5].

Other studies on the bacteriology of AFs and abscesses have increasingly shifted their attention towards inflammatory rather than infectious mechanisms [6,7]. Our previous study confirmed that the expression of cytokines IL-1beta and IL-8 could play a role in the persistence of AFs, and also demonstrated the potential importance of the epithelium–mesenchymal transition (EMT) in the pathogenesis of the disease [8].

The human amniotic membrane (hAM) is the innermost layer of the amniotic sac in which the fetus develops; it is a thin non-vascularized sheet in which two layers, epithelial and stromal, can be distinguished, and from which it is possible to isolate human amniotic epithelial cells (hAEC) and human amniotic mesenchymal stromal cells (hAMSC) [9]. 

The hAM has a long history of clinical applications [10]. The first documented use of the hAM as a surgical material was for skin transplantation [11]. Since then, the hAM has been widely used in clinical practice, such as in ophthalmology, for corneal epithelial defects and ulcers, glaucoma, pterygium, and bullous keratopathy [12,13,14,15]. Furthermore, the hAM is being employed in dermatology to cover burns and to treat chronic ulcers [16]. The many current clinical applications of the hAM corroborate its safety [10]. 

Several in vitro and preclinical studies clarified that the mechanism of action through which the hAM and hAM-derived cells contribute to tissue repair and regeneration is through their immunomodulatory properties and by promoting the resolution of inflammation consequent to injury [9].

Our hypothesis was that the hAM could modulate the inflammatory condition related to the persistence of AFs, leading to normal wound healing. In this report we describe for the first time the implant of hAM in a patient affected by cryptoglandular AF.

## 2. Materials and Methods

A human amniotic membrane patch was provided by Fondazione Banca dei tessuti di Treviso Onlus (Treviso, Italy), an Italian human tissue bank in charge of procuring, processing, storing and distributing tissues for transplant in humans. The hAM patch, a very thin transparent resistant sheet was provided cryopreserved with the size 3 × 3 cm [17]. The hAM was cryopreserved with 10% DMSO as a cryoprotective agent. 

A female patient (38 years old), with no comorbidities, was referred to our Center after ambulatory drainage of a perianal abscess which occurred 6 months earlier. The patient underwent evaluation under anesthesia, with a curettage of the residual abscess cavity and placement of a loose seton for drainage, maintained for 70 days up to the operation shown in this article. After 3 months, the patient was evaluated by 3D-endoanal ultrasound (3D-EAUS, model 2202, BK Medical, Herlev, Denmark), and treated surgically after signing a written informed consent. The 3D-EAUS demonstrated the resolution of sepsis but the persistence of a left anterolateral transphincteric fistula with the internal orifice located at the middle anal canal and the external orifice at the left anterolateral perianal area, 2 cm from the anal verge (Figure 1).

### Surgical Procedure

In the morning of surgery, two enemas were performed as bowel preparation. Antibiotics (ciprofloxacin 400 mg + metronidazole 500 mg) were administered preoperatively. Under general anesthesia, the patient was placed in the lithotomy position. 

Following an accurate curettage of the AF with a brush and Volkmann spoon to remove both the intraluminal epithelium and granulation tissue, the fistula tract was cannulated with a fistula probe and prepared for the implant of the hAM (Figure 2). 

The cryopreserved hAM resting on a filter was then thawed in the operating room by immersion in a bath of saline solution at 40 °C (Figure 3). 

It was subsequently washed twice in saline solution at 25 °C; finally, one side of the square hAM was transfixed with a resorbable suture (3-0 Vicryl™, Ethicon Endo-Surgery, Inc., Cincinnati, OH, USA), thus creating an implantable fusiform patch (Figure 4). Attention was paid to place the epithelial side of the hAM outward, to face the lumen of the fistula tract.

Following the placement of a 3-0 Vicryl suture inside the AF tract, the hAM was carefully implanted into the fistula tract, pulling it upward from the external to the internal opening (Figure 5). 

The inner and the outer parts of the membrane were then sutured to the internal and external fistula opening by a 3-0 Vicryl suture (Figure 6). The internal fistula opening was closed with a z-shaped stitch of 3-0 Vicryl suture. The external fistula opening was left open to permit serum drainage.

## 3. Results

No intraoperative or postoperative complications occurred. The patient was discharged the day after the procedure f an uneventful hospitalization; she suffered no pain. Stool softeners and analgesics were prescribed as needed. A resting period of one week and a diet rich in water and fiber were also prescribed.

At the 1-week follow-up visit no pain (VAS 0) was referred by the patient, no inflammation was evident at the level of the previous external fistula opening (Figure 7). The same findings were observed at the 1-month and the 3-months follow-up visits.

## 4. Discussion

Still today, none of the surgical techniques proposed for the treatment of anal fistulas are considered the “gold-standard” [18].

The more accredited theory on the etiology of perianal fistulas dates back to 1961, and considers infection of the anal canal glands as the initial source of the problem [19]. Since then, no other significant advances have been established in the field, even if it has been shown that the chronic phase of the disease is more related to a dysregulated inflammatory process rather than an infectious condition [6,7]. In 2013, van Onkelen et al. demonstrated that in 9 out of 10 excised anal fistulas there were traces of peptidoglycan, a component of the bacterial wall, hypothesizing that it could be related to the secretion of pro-inflammatory cytokines [20].

Acute, self-limiting and resolutive inflammation is of crucial importance in the repair processes of injured tissues. Conversely, chronic inflammation can lead to excessive tissue damage and deregulated tissue healing. Amniotic cells and derivatives can participate in the resolution of inflammation by acting on various inflammatory mediators such as cytokines and chemokines [9]. In the context of tissue repair, a decisive role is played by macrophages, which are generally classified as M1 or M2 with reference to their different pro- or anti-inflammatory activity, mediated by the secretion of different cytokines. The administration of amniotic cells was shown to modulate the balance between M1 and M2 macrophages along the process of tissue repair [21]. We have previously demonstrated that mesenchymal stromal cells from the amniotic membrane, and their secretome, are able to induce macrophage polarization from inflammatory (M1) to anti-inflammatory (M2) [22]. However, recently it has become increasingly evident that decellularized membrane and derived products can also stimulate tissue repair. For example, decellularized amniotic membrane was shown to support human fetal fibroblast cell proliferation in vitro [23]. In addition, others have demonstrated how the use of decellularized amniotic membrane was able to modulate the inflammatory response of immune cells and to induce polarization towards anti-inflammatory M2-macrophages [24]. At the same time, the application of freeze-dried amniotic membrane or hydrogels derived from the membrane were able to promote the formation of a mature epidermis and dermis-with composition similar to healthy skin-in a porcine wound healing animal model [25]. Finally, the mechanism underlying the ability of the amniotic membrane to mechanically limit the inflammatory response has also been reported. In fact, the authors reported how the ocular transplantation of amniotic membrane patches exhibited an ability to trap immune cells and how these were induced to undergo apoptosis [26]. All of this evidence supports the fact that the amniotic membrane, even if devoid of viable cells, can still retain the potential to exert pro-regenerative and potentially immunomodulatory actions.

For these reasons, the hAM has already been widely employed in delayed wound healing, with encouraging results [27].

Our hypothesis is that the implantation of the hAM in AFs can modulate underlying imbalanced inflammation thus facilitating the reparative process, leading to healing. To our knowledge, this is the first report of the safe implantation of the hAM for the treatment of an AF in a human patient, without intra- or post-operative complications; this should be almost obvious since hAM application represents a “minimally-invasive” procedure that fully preserves the sphincter function.

In a study by Rafati et al., 14 rabbits with high AFs were randomly treated using endorectal flap or endorectal flap + a 1 × 1 cm patch of hAM: post-surgical pathological evaluation revealed increased collagenization, neovascularization, epithelialization, as well as less tissue necrosis, inflammation, and granulation tissue formation in the group treated with the hAM. Moreover, clinical wound healing occurred more rapidly in the hAM group [28]. Similarly, it has been used successfully in an animal model of rectovaginal fistula [29], and to prevent anastomotic leak after colonic resections in rats [30].

A recent pilot study applied the hAM to prevent the formation of urethra-cutaneous fistulas and other complications in 28 patients with a previous failed hypospadias repair: urethra-cutaneous fistula occurred in only two cases [31].

An ongoing study at our institution will analyze the clinical efficacy of this technique in the treatment of cryptoglandular AFs in a larger group of patients to be observed in the long-term follow-up. Successful results in the preliminary application of hAM in AFs could allow its wider use, especially for complex cases where sphincter cutting surgery would lead to a high risk of continence impairment or where a sphincter-saving procedure would produce a high recurrence rate.

## 5. Conclusions

The implant of hAM in a female patient affected by cryptoglandular AF was safely and easily performed. Moreover, future studies to assess the safety in the long-term follow-up, and its clinical efficacy, are needed.

## Figures and Tables

**Figure 1 jcm-11-01350-f001:**
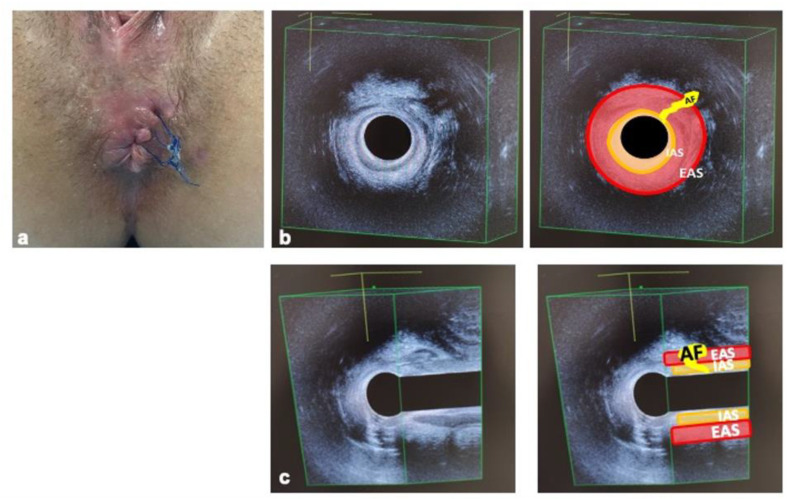
Preoperative aspect (**a**), and 3D-EAUS imaging ((**b**), axial view; (**c**), longitudinal view) of the transphinteric anal fistula to be submitted to the human amniotic membrane implantation. Fistula resulted from an anal abscess previously drained with seton placement. AF: anal fistula; IAS: internal anal sphincter; EAS: external anal sphincter.

**Figure 2 jcm-11-01350-f002:**
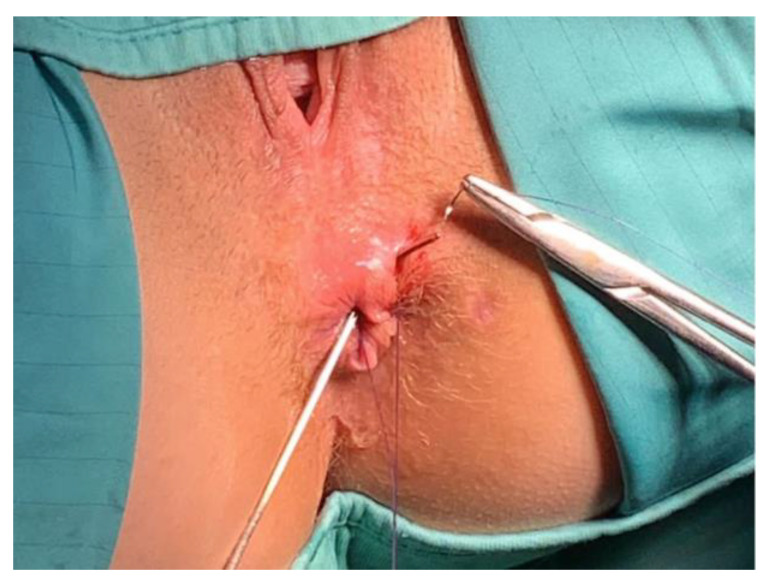
Following an accurate curettage of the AF with a brush and Volkmann spoon removing both the intraluminal epithelium and granulation tissue, the fistula tract was cannulated with a fistula probe and prepared for the implant of the hAM.

**Figure 3 jcm-11-01350-f003:**
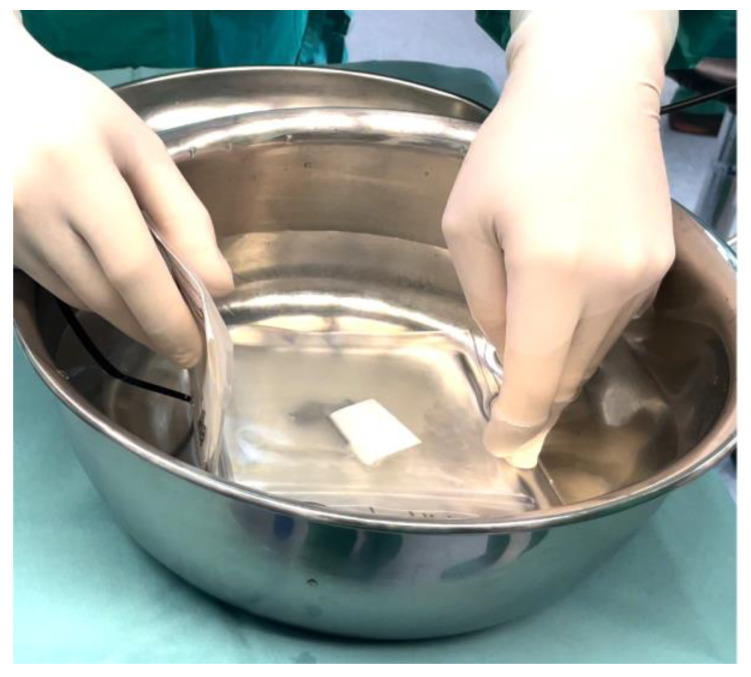
The cryopreserved hAM, placed on a filter, was thawed by immersion in a bath of saline solution at 40 °C.

**Figure 4 jcm-11-01350-f004:**
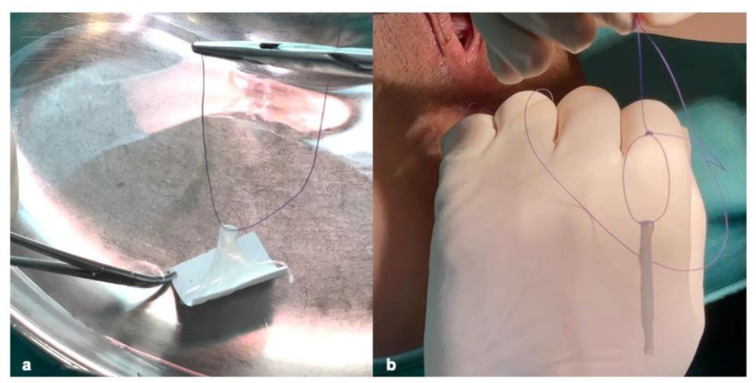
(**a**) The square hAM was transfixed with a resorbable 3-0 Vicryl™ suture (Ethicon Endo-Surgery, Inc., Cincinnati, OH, USA), (**b**) thus creating an implantable fusiform patch (**b**).

**Figure 5 jcm-11-01350-f005:**
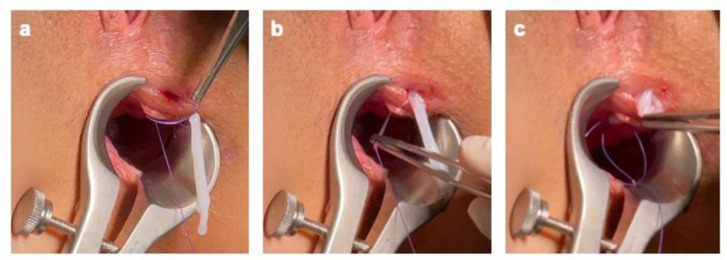
(**a**–**c**) The hAM was carefully implanted into the fistula tract, pulling up it from the external to the internal opening.

**Figure 6 jcm-11-01350-f006:**
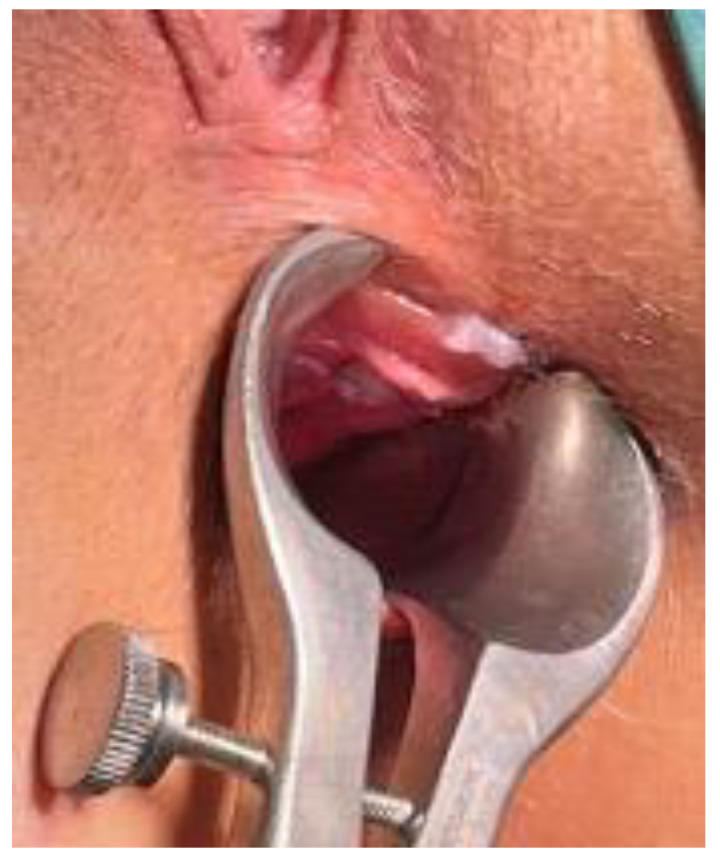
The inner and the outer parts of the membrane were sutured to the internal and external fistula opening by a 3-0 Vicryl suture, preventing its dislocation.

**Figure 7 jcm-11-01350-f007:**
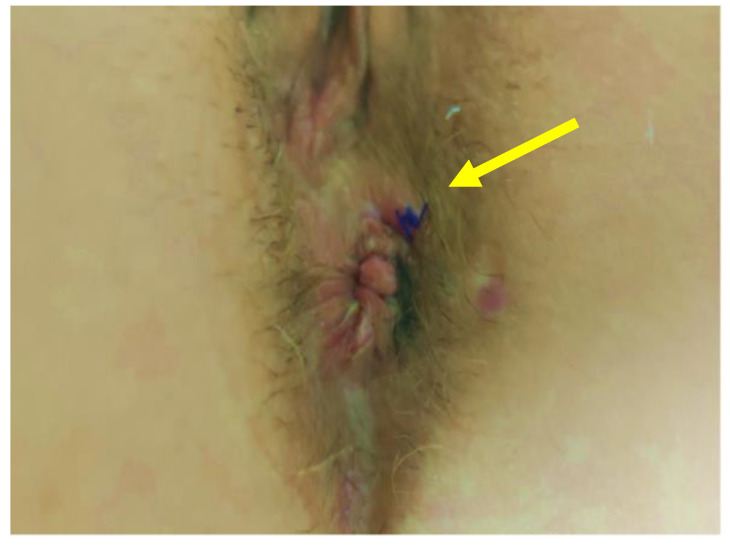
View of the perianal area 7 days after the hAM implant: no inflammation at the previous external anal fistula opening (arrow).

## Data Availability

The data presented in this study are available on reasonable request from the corresponding author.

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
