# Peer review of "Human Amniotic Membrane for the Treatment of Cryptoglandular Anal Fistulas"

_jcm, 2022, doi:10.3390/jcm11051350_

Round 1

Reviewer 1 Report

With the Treatment of Cryptoglandular Anal Fistulas, the authors describe another valuable and successful application of hAM.

The manuscript is clearly structured and written in a way that is easy to understand. However, a few additional information should be added before it is accepted for publication.

Page 2, lines 67ff          please specify the characteristics of the used hAM. Was it cryopreserved with a cryo protective agent? If yes, what CPA was used? In which concentration? The cited reference (EDQM-Guide) does not contain the information about the processing and preservation of the membrane used here.

Page 3, line 99              why was the membrane washed? Did a CPA have to be removed? However, factors are also lost in the process.

Page 3/4, lines 100ff     was attention paid to which side of the hAM (epithelial or stromal side) was facing outward and which was curled up inside the fusiform patch?

Page 5, lines 146ff        in the discussion it is raised that the administration of amniotic cells has been shown to modulate the balance between M1 and M2 macrophages along the process of tissue repair [21,22]. Do the authors assume that vital cells were necessary or responsible for the successful healing of the fistula? Please discuss this point more.

Author Response

With the Treatment of Cryptoglandular Anal Fistulas, the authors describe another valuable and successful application of hAM.

  • Answer: thanks for the comment.

The manuscript is clearly structured and written in a way that is easy to understand. However, a few additional information should be added before it is accepted for publication.

  • Answer: thanks for the comment.

Page 2, lines 67ff  please specify the characteristics of the used hAM. Was it cryopreserved with a cryo protective agent? If yes, what CPA was used? In which concentration? The cited reference (EDQM-Guide) does not contain the information about the processing and preservation of the membrane used here.

  • Answer: thanks for the comment. Human amniotic membrane is cryopreserved with 10% DMSO as a cryoprotective agent. We have added a sentence to the manuscript to state it.

Page 3, line 99  why was the membrane washed? Did a CPA have to be removed? However, factors are also lost in the process.

  • Answer: thanks for the comment. After thawing, hAM is washed twice in sterile saline solution before implantation, to remove CPA. However, after washing hAM still retains cytokines as published by Paolin et al. (Paolin A, et al. Cytokine expression and ultrastructural alterations in fresh-frozen, freeze-dried and γ-irradiated human amniotic membranes. Cell Tissue Bank. 2016 Sep;17(3):399-406).

Page 3/4, lines 100ff     was attention paid to which side of the hAM (epithelial or stromal side) was facing outward and which was curled up inside the fusiform patch?

  • Answer: thanks for the comment. Attention was paid to put the epithelial side of the hAM outward, to face the lumen of the fistula tract. We have added a sentence to the manuscript to state it.

Page 5, lines 146ff        in the discussion it is raised that the administration of amniotic cells has been shown to modulate the balance between M1 and M2 macrophages along the process of tissue repair [21,22]. Do the authors assume that vital cells were necessary or responsible for the successful healing of the fistula? Please discuss this point more.

  • Answer: we thank the reviewer for this observation. We have previously demonstrated that mesenchymal stromal cells from the amniotic membrane, and their secretome, are able to induce macrophage polarization from inflammatory (M1) to anti-inflammatory (M2) (Magatti M, et al. J Tissue Eng Regen Med. 2017;11:2895-2911. doi: 10.1002/term.2193). However, recently it has become more evident that decellularized membrane and derived products can also stimulate tissue repair. For example, decellularized amniotic membrane has been shown to support human fetal fibroblast cell proliferation in vitro (10.22074/cellj.2015.493.). In addition, others have demonstrated how the use of decellularized amniotic membrane was able to modulate the inflammatory response of immune cells and to induce polarization towards anti-inflammatory M2-macrophages (10.3390/ijms19041032). At the same time, the application of freeze-dried amniotic membrane or hydrogels derived from the membrane were able to promote the formation of a mature epidermis and dermis - with composition similar to healthy skin - in a porcine wound healing animal model (10.1002/sctm.19-0101). Finally, it has also been reported how the amniotic membrane was able to mechanically limit the inflammatory response. In fact, the authors reported how the ocular transplantation of amniotic membrane patches was able to trap immune cells and how these were induced to undergo apoptosis (10.1097/00003226-200105000-00015).  All this evidence supports the fact that the amniotic membrane, even if devoid of viable cells, can still retain the potential to exert pro-regenerative and potentially immunomodulatory actions. We have added this sentences in the Discussion section.

Reviewer 2 Report

  1. The manuscript is well written. A new treatment concept has been described in a single case of low simple patient.
  2. Strength: A novel technique has been described. Weakness: Only 1 case that too a low, simple fistula with tract length quite small.
  3. The main concern is that there is only one case and in that case too, the fistula tract length is hardly 1 cm. For such simple low fistulas, fistulotomy has 95-98% success rate without any risk of incontinence. The purpose of developing new procedures is to find cure for high complex fistulas while preserving the sphincter-complex. Therefore, I would request to try this procedure in a couple of high, complex fistulas and if it is effective in those cases, then this procedure merits publication and further research.
  4. The follow-up is not long enough. It is preferable to have a longer follow-up to ascertain the success rate of a procedure.

The manuscript is well written. However, the main concern is that there is only one case and in that case too, the fistula tract length is hardly 1 cm. For such simple low fistulas, fistulotomy has 95-98% success rate without any risk of incontinence.

The purpose of developing new procedures is to find cure for high complex fistulas while preserving the sphincter-complex. Therefore, I would request to try this procedure in a couple of high, complex fistulas and if it is effective in those cases, then this procedure merits publication and further research.

Author Response

1. The manuscript is well written. A new treatment concept has been described in a single case of low simple patient.

  • Answer: thanks for the comment.

2. Strength: A novel technique has been described. Weakness: Only 1 case that too a low, simple fistula with tract length quite small.

  • Answer: thanks for the comment. The main purpose of this technical note was to describe for the first time the technical details and the novelty of the procedure.

3. The main concern is that there is only one case and in that case too, the fistula tract length is hardly 1 cm. For such simple low fistulas, fistulotomy has 95-98% success rate without any risk of incontinence. The purpose of developing new procedures is to find cure for high complex fistulas while preserving the sphincter-complex. Therefore, I would request to try this procedure in a couple of high, complex fistulas and if it is effective in those cases, then this procedure merits publication and further research.

4. The follow-up is not long enough. It is preferable to have a longer follow-up to ascertain the success rate of a procedure.

  • Answer to point 3 and 4: thanks for the comments and suggestions. Again, the main purpose of the paper was the description of surgical procedure of hAM implant. Of course, the single case described in the paper has no relevance about both the “general efficacy” of this approach and the outcome of that single patient, due to the short-term follow up. We are fully convinced that this procedure could be a good option to treat patients with very complex fistulas, because high or located anterior in female patients (where a simple fistulotomy could be detrimental for fecal continence). In fact, a specifically addressed prospective clinical study is ongoing in our Centre and we will be very glad to publish the results as soon as the patients’ accrual will be completed, and the follow up period will be long enough.

Round 2

Reviewer 2 Report

I am satisfied with the explanation.